# PROCEEDINGS A

mathematics

epidemic model, Euler–Lotka equation, prevention, estimation, face mask

**Author for correspondence:**
Tom Britton
e-mail: tom.britton@math.su.se

# Quantifying the preventive effect of wearing face masks

Tom Britton

Department of Mathematics, Stockholm University, Stockholm, Sweden

TB, 0000-0002-9228-7357

An important task in combating the current Covid-19 pandemic lies in estimating the effect of different preventive measures. Here, we focus on the preventive effect of enforcing the use of face masks. Several publications study this effect, however, often using different measures such as: the relative attack rate in case-control studies, the effect on incidence growth/decline in a specific time frame and the effect on the number of infected in a given time frame. These measures all depend on community-specific features and are hence not easily transferred to other community settings. We argue that a more universal measure is the relative reduction in the reproduction number, which we call the *face mask effect*, $E_{FM}$. It is shown how to convert the other measures to $E_{FM}$. We also apply the methodology to four empirical studies using different effect-measures. When converted to estimates of $E_{FM}$, all estimates lie between 15 and 40%, suggesting that mandatory face masks reduce the reproduction number by an amount in this range, when compared with no individuals wearing face masks.

## 1. Introduction

A main motivation for modelling and statistical analyses of infectious disease outbreaks is to understand and to estimate the effect of introducing different preventive measures. Prior to Covid-19, the type of preventive measure that by far had received the most attention in this regard is vaccination. There are, for example, recipes for estimating the vaccine efficacy with respect to susceptibility, infectivity, disease/symptoms, overall and more, under various settings (e.g. [1,2]).

Here our focus lies on quantifying the effect of wearing face masks in the current Covid-19 pandemic. More specifically, we aim at quantifying the effect of making face masks mandatory in a community compared with no individuals wearing face masks. Several studies from different regions estimate the effect of face mask regulations but most of them use efficacy measures that are specific to the region of interest and hence are hard to transfer to other settings.

The purpose of the present paper is to define a face mask effect being directly linked to the reduced risk of getting infected when wearing a face mask *and* to the reduced risk of infecting others when wearing a face mask, thus making it more easily transferable to other community settings. Using modelling results from epidemic theory, we describe how to derive estimates of this effect from other community-specific efficacy measures. We apply the methodology to four studies, with data from different regions and that use different types of data to quantify the preventive effect of face masks.

## 2. The face mask effect $E_{FM}$

When it comes to effects of wearing face masks, the two most direct effects are the efficacy with respect to susceptibility $e_S$, and the efficacy with respect to infectivity $e_I$. By $e_S$ we mean the reduced risk of infection in a contact between an infective without a face mask and a susceptible wearing a face mask, as compared with the susceptible not wearing a face mask, so $1 - e_S = P(i_{No} \to s_{FM})/P(i_{No} \to s_{No})$, where '$\to$' means an infection occurs in a contact and the index on $i$ and $s$ reflect whether the susceptible/infective wears a face mask or not. Similarly, by $e_I$ we mean the reduced risk of infection in a contact with a susceptible not wearing a face mask when the infective wears a face mask compared with when the infective does not wear a face mask, so $1 - e_I = P(i_{FM} \to s_{No})/P(i_{No} \to s_{No})$.

Furthermore, we make the natural assumption that these two effects act multiplicatively, so that the reduced risk of infection if both individuals wear face masks compared with when neither of them do is given by $1 - P(i_{FM} \to s_{FM})/P(i_{No} \to s_{No}) = 1 - (1 - e_S)(1 - e_I)$. Since all transmission probabilities are reduced by this factor when all individuals go from not wearing a face mask to all wearing face masks, it follows that this effect will also be the relative reduction in the reproduction number when switching from a community with no one wearing a face mask (having reproduction number $R_{No}$) to all individuals wearing a face mask (with reproduction number $R_{FM}$). We call this the face mask effect $E_{FM}$:

$$E_{FM} = 1 - \frac{R_{FM}}{R_{No}} = 1 - (1 - e_S)(1 - e_I). \tag{2.1}$$

It is important to note that this effect is independent of the underlying reproduction number $R_{No}$, which typically depends on which community is studied, what other preventive measures are currently in force and how much immunity the community currently has.

Here, we are considering the effect of the preventive measure to wear a face mask. The same methodology can of course be applied to other individual preventive measures that reduce susceptibility and/or infectivity, such as vaccination, extended hand-washing and more.

## 3. Estimation of $E_{FM}$ from different empirical data sources

We now make use of the modelling results in the previous section to obtain estimates of the face mask effect $E_{FM}$ from empirical studies estimating other measures of the effect of wearing a face mask.

### (a) Estimating $E_{FM}$ from a randomized control trial

Bundgaard *et al.* [3] was based on a trial in Denmark in which approximately 3000 randomly selected individuals were instructed to follow all recommendations *and* to wear a face mask, and another 3000 individuals were instructed to follow all recommendations but to *not* wear a face

mask. All individuals were susceptible at the start of the study period on 3 April, and after 2 months they were tested for infection. There were some drop-outs during the study, so the final result was that 42 out of 2392 assigned to wear face masks were infected, and in the other group 53 out of 2470 got infected. The (small) risk of getting infected during the study period was hence 1.8% (42/2390) for individuals wearing a face mask, and 2.1% (53/2470) for the control group. The reduced risk of getting infected in case of wearing a face mask when compared with not wearing a face mask is hence estimated to equal 18% (= 1 − 0.018/0.021). The two groups were by no means separated in the community, so the reduced risk is attributed to reduction in susceptibility; the potential reduction in infectivity (if infected) cannot be estimated from such a study.

The study has two important advantages. The first is that individuals were randomly distributed to the two groups, thus making the risk for confounding effects negligible. The other advantage (which is also a disadvantage!) was that the risk of getting infected was quite small. As a consequence, since the vast majority did not get infected, it is reasonable to interpret the reduction by 18% as a reduction effect *per contact*, so that $e_S$ can be estimated $\hat{e}_S = 0.18$. If a bigger fraction would have been infected, individuals would have been exposed to infections many times, and there would be no reason to expect the fraction infected among individuals wearing a face mask to be $e_S$ lower than among individuals not wearing a face mask.

Because the total number infected in both arms were small the estimated effect of $\hat{e}_s = 18\%$ has a wide confidence interval spanning from −23% up to +46% (a negative effect would imply that wearing a face mask *increases* the risk of getting infected), and the study does hence not show that the reduction in susceptibility is statistically significant.

Still, the point estimate $\hat{e}_S = 0.18$ indicates a protective effect for the risk of getting infected, but the study gives no information on the reduction in infectivity $e_I$. Experimental studies (e.g. [4]) indicate that $e_I > e_S$, so a conservative estimate of the overall preventive effect of wearing face masks would be to assume $e_I = e_S$, which gives the estimated face mask effect

$$\hat{E}_{\mathrm{FM}} = 1 - (1 - 0.18)^2 = 0.33.$$

This estimate is of course equipped with high uncertainty, but the point estimate at least suggests that the reproduction number is reduced by approximately 33%. If we instead assume that the reduction in infectivity is 50% higher than that of susceptibility, then the estimate of $E_{\mathrm{FM}}$ would be $1 - (1 - 0.18)(1 - 0.27) = 0.40$, an even bigger effect.

## (b) Estimating $E_{\mathrm{FM}}$ from incidence change in a comparative study

A more common type of study when estimating the face mask effect are retrospective studies. A potential problem with these type of studies is of course that even if attempts are made to remove potential biases, there is always a risk that the communities compared or the observed time periods differ in other aspects than the use of face masks.

One such study [5] compared different counties in Kansas, USA, some of which had enforced face mask regulations and others that did not. The regulations were put into force on 5 July 2020, and it was studied how the incidence grew before the change and after the change, and these changes in growth rate were compared with changes in growth rate for the same periods in counties that did not enforce the wearing of face masks. We postpone estimation from comparing different regions to the next subsection (for another study) and focus here on the change of incidence over time before and after the introduction of face mask regulations.

The study showed that in counties that were later in having mandatory face masks the average incidence in early June was 3, in early July it had risen to 17 (after which regulations were put in place), and in mid-August the incidence rate was 16, so the incidence rate stopped growing after face mask regulations were put in place.

The study reports that incidence increased by 0.25 cases per 100 000 individuals per day prior to face mask regulations whereas incidence *decreased* by 0.08 cases per 100 000 individuals per day after face masks became mandatory. Even though these numbers are completely true they carry very little information on what might be the effect in a different community. For this reason, we

now estimate the face mask effect $E_{FM}$ from the same data, using the methods described in the modelling section.

Since communities without face mask regulations had no drop in exponential growth rate (in fact a small increase), we attribute the change in incidence in communities adopting face mask regulations entirely to this effect (as in [5]).

The incidence was reported to equal 3.0 cases per 100 000 individuals on 3 June (weekly average with 3 June as midpoint) and incidence as 17.0 cases per 100 000 individuals on 5 July, 32 days later. Using (A 3) in the appendix with $d = 32$, we obtain an estimate of the exponential growth prior to face mask regulations as $\hat{r}_{No} = \ln(17.0/3.0)/32 = 0.054$. In order to estimate $R_{No}$ (and eventually $E_{FM}$) using (A 1) from the appendix, we need to make some assumptions about the generation time and we choose $\mu = 6.5$ days and $\sigma = 4$ days as in [6]. By inverting (A 1), we obtain the estimate $\hat{R}_{No} = 1.39$.

The growth rate after regulations was more or less nil, or in fact a very small decline (from 17 to 16 in 45 days). This suggests a reproduction number very close to 1; using the same methodology we get $\hat{R}^{(FM)} = 0.99$. The estimate of the overall effect of wearing face mask from the Kansas study is hence

$$\hat{E}_{FM} = 1 - \frac{\hat{R}_{FM}}{\hat{R}_{No}} = 0.29.$$

The Kansas study [5] hence suggests that the effect of enforcing face masks to be worn has the overall preventive effect $\hat{E}_{FM} = 29\%$. This estimate is equipped with less uncertainty than the Danish study but may on the other hand have confounding factors resulting in a biased estimate. For instance, a difference between counties imposing face mask regulations early and counties that did not was that the former experienced a quicker epidemic growth in June. Most likely this partly explains their willingness to enforce face masks, but there is of course also a possibility that individuals in these communities became more precautious compared with individuals in communities that experienced lower growth rates. If this was the case, the estimated effect should only partly be attributed to face masks.

## (c) Estimating $E_{FM}$ from the change in the number of infections in a comparative study

Another recent comparative study [7] analyses the effect that an early face mask regulation had in the German city of Jena, as compared with a 'synthetic control group' defined with the aim to minimize other confounding effects. One of the main results was that the number of reported cases (per 100 000 individuals) during a 20-day time window dropped by 75% when comparing Jena with the synthetic control group. More precisely, the cumulative number of reported cases in Jena on 6 April when the face mask regulation was introduced was 142 (per 100 000 individuals), and 158 reported cases on 26 April. The corresponding numbers for the synthetic control group were 143 and 205. The number of infections in the time window was hence 16.0 in Jena and 63.0 in the synthetic control group (per 100 000 individuals), a drop of 75% in Jena compared with the synthetic control group. In order to estimate the epidemic growth, we also needed to know the number of infectious people at the start of the study period (a low number would suggest a quick growth and the opposite if the number of infectious people was high). This information is of course harder to obtain, but if we assume a latent period of 3 days followed by an infectious period of 4 days (in line with [6]) then this number can be estimated from the reported number of cases prior to 6 April. These numbers are close to identical for Jena and the control group and, since the number of reported cases is very close to 4 per day (per 100 000 individuals), we estimate the number of infectious individuals on 6 April to be 16 for both Jena and the synthetic control group.

Using equation (A 4) of the appendix, we can now estimate $r_{FM}$ for Jena and $r_{No}$ for the synthetic control group. For Jena the left-hand side equals 16/16 and the right-hand side has $r$ as unknown and $\mu = 6.5$ and $d = 20$ (the length of the time window) giving the estimate $\hat{r}_{FM} = -0.065$, and for the synthetic control group the right-hand side is the same but the left-hand side is 63/16, giving the estimate $\hat{r}_{No} = 0.015$. Using (A 2) in the appendix, with $\mu = 6.5$ and

**Table 1.** Estimates of the face mask effect $E_{FM}$ from different empirical studies.

| reference/country | type of reported data | submodel | estimated $E_{FM}$ |
|---|---|---|---|
| [3] Denmark | relative attack rate | $e_I = e_S$ | 33% |
| [3] Denmark | relative attack rate | $e_I = 1.5e_S$ | 40% |
| [5] USA | change of incidence | | 29% |
| [7] Germany | change in reported cases | Jena only | 43% |
| [7] Germany | change in reported cases | combined estimates | 21% |
| [8] 190 countries | $E_{FM}$ | | 15% |

$c = 4/6.5 = 0.615$, we get $\hat{R}_{FM} = 0.63$ for Jena and $\hat{R}_{No} = 1.10$ for the synthetic group, thus resulting in an estimate of $E_{FM}$ of $\hat{E}_{FM} = 1 - \hat{R}_{FM}/\hat{R}_{No} = 43\%$.

This estimate of the face mask effect $E_{FM}$ is notably higher than the other two estimates of $E_{FM}$, which both are close to 30%. However, Mitze *et al.* [7] argue that the decrease in number of infections by 75% might be an over-estimate and when taking other aspects into account (including the effect of face mask regulations in other regions later on) suggest a better estimate of the reduction in the number of infections attributed to face mask regulations as 45%, which would correspond to 34.6 infections (per 100 000 inhabitants) rather than 16. If instead this number was used in the estimation procedure, the face mask effect would be estimated as $\hat{E}_{FM} = 21\%$.

## (d) Explicit estimate of $E_{FM}$ from meta analyses

We have found one recent study that estimates the same efficacy measure $E_{FM}$ as suggested here [8]. This study conducts a type of meta analysis using data from 190 countries and estimates the effects of several NPIs. Their estimate equals $\hat{E}_{FM} = 15.1\%$ when adjusting for confounding factors (the estimate equals 33% before this correction). Needless to say, there is always a risk of missing important features when analysing this many countries simultaneously. For example, the adherence may be different in different countries and there is always a risk for confounders, such as changed behaviour without new regulations. The estimated effect 15.1% is hence an average over different countries and possible confounders.

## 4. Conclusion and discussion

Using results from mathematical theory for epidemic models we relate different observable quantities with the current reproduction number $R$. By inverting these relations it is possible to obtain and compare estimates of the reproduction number with and without face masks from empirical studies investigating such quantities. Four studies were used to obtain estimates of the face mask effect $E_{FM} = 1 - R_{FM}/R_{No}$, where $R_{FM}$ is the reproduction number *with* enforced face masks, and $R_{No}$ is the reproduction number when individuals are not using face masks. In table 1, we summarize our results. The Danish study [3] results in two estimates of $E_{FM}$ depending on if the reduced risk of infecting others when wearing a face mask, $e_I$, is equal or 50% higher than the reduced risk of getting infected by wearing a face mask $e_S$. Also, the German study [7] has two estimates of $E_{FM}$ depending on if only Jena is used for estimating $E_{FM}$ or if also estimates from other cities are used (as suggested in [7]).

As expected, the different estimates show some variation. Still, all suggest that the effect of introducing face mask regulation reduces the reproduction number by 15–40%, a rather big reduction for one single preventive measure. Even at the lower end of this interval ($E_{FM} = 15\%$), a community not using face masks and currently having $R_{No} \approx 1.2$ (with a doubling time of daily cases being less than a month) would change its reproduction number to $R_{FM} \approx 1$ if face masks

were made mandatory, hence more or less stopping the growth. If instead $E_{FM} = 25\%$ the new reproduction would be $R_{FM} \approx 0.9$ and transmission would start declining in the community.

The estimates above are estimates when going from no people wearing face masks to all individuals wearing face masks. If there are in some community already a fraction $f_B$ wearing face masks before the regulation is put in place, and the recommendation/regulation results in that a higher fraction $f_A$ wear face masks (but not necessarily all individuals), then an estimate of this effect would be $(f_A - f_B)E_{FM}$, so if for example a recommendation results in that the fraction wearing face masks increases from 20% to 70%, then the effect would be $0.5 * E_{FM}$.

All four studies may contain errors. The Danish study [3] is randomized thus minimizing potential confounders, but on the other hand few individuals were infected in each group thus leading to rather high uncertainty. In fact, no effect ($E_{FM} = 0$) is contained in the confidence interval. The German [7] and Kansas [5] studies contain more infected individuals and hence have less uncertainty, but here the two compared groups come from different regions. These different regions may differ also in other aspects, for example regions imposing face mask regulations may have higher incidence, making individuals more preventive in general, and the estimated face mask effect will then include this additional preventive behaviour. Furthermore, the communities may have changed behaviour with regards to disease spreading during the study period, which we have assumed not to be the case. This can clearly have an effect on the estimates, and if these behaviour changes during the study period differ between regions it will affect the estimate of $E_{FM}$ more. Finally, the meta-analysis study [8] compares the effect of face mask regulations in 190 countries, some with regulations and others without. Clearly, these countries may differ very much also in other aspects.

All four studies indicate a positive effect of introducing face mask regulations. There is of course a risk of publication bias in that there might exist other studies that show no significant effect of face masks, but that these studies never reached publication. However, the Danish study [3] can be claimed to belong to this category. There the reduction in susceptibility was estimated as 18% but this estimated positive effect was not significant, and the most common reaction in the media was that face masks had no significant effect.

The four studies estimate the effect of going from no face masks to all individuals wearing face masks. This might hold in the randomized Danish study, but in the other studies there were surely individuals in the control regions who wore a face mask (although the studies are from the first wave when general use of face masks was much lower then during later waves). Similarly, most likely there were individuals in the regulated regions who did not adhere to the regulations. The estimates of $E_{FM}$ are then not estimates of the effect of going from no use to complete use of face masks, but rather going from low to high use of face masks. In this case, the true effect of going from no use to complete use would be slightly higher than the estimates in table 1.

Even if the suggested measure $E_{FM}$ is claimed to be 'universal' this is not entirely true. For example, if in one community people adhere strictly to the face mask regulation, also outdoor and in schools, but in another community people do not wear face masks outdoors or in schools, then the face mask effect will of course be higher in the former community. The individual face mask efficacy also depends on: the type of face mask, whether or not the face mask is worn properly, if it is removed and put on in a safe way, and how well it fits the face of the user. Similarly, the reduced effect may differ between different types of contact, the reduction might e.g. be higher for quick contacts and slightly smaller for longer face to face meetings. However, unless different communities are systematically different in these aspects, the suggested measure $E_{FM}$ is a population average effect that should not vary greatly between communities, and its estimation is important to increase understanding of how to prevent/reduce transmission.

Data accessibility. This article has no additional data.

Competing interests. We declare we have no competing interests.

Funding. The author is grateful for financial support from NordForsk, grant no. 105572.

# Appendix A. Mathematical background and assumptions

In the appendix, we express the effect of introducing face mask regulation on other quantities in an ongoing outbreak, using theory for epidemic models (e.g. [9]). But first we specify our assumptions.

## (a) Underlying assumptions

In our analyses, we assume that other aspects of the epidemic except for face mask regulations, such as preventive measures, immunity and seasonal effects, remain unchanged during the study period. Of course, this is rarely completely true but if the study period lasts for 1 month or 2, as in the studies considered here, it might at least hold approximately.

   We stress that the theory is not only applicable for the initial stages of an outbreak but also later in the epidemic, as long as preventive measures and immunity remain stable during the study period, implying that we have a stable reproduction number $R$ over the study period. Similarly, the community under consideration is allowed to contain heterogeneities. The assumption we make is that such heterogeneities do not alter the spreading over the study period. For instance, younger age groups may be more socially active and hence have a higher risk of getting infected and infecting others. What we then assume is that the fraction of susceptibles in the different age groups does not change much during the study period, something which approximately holds true during 1 or 2 months.

## (b) Effect on the exponential growth/decline rate

For an epidemic that currently has $I(t)$ infectious individuals and current reproduction number $R$ (be it with or without mandatory face masks), is known to progress exponentially

$$I(t+s) = I(t)e^{rs},$$

where $r$ is known as the Malthusian parameter. This exponential growth (or decay) depends on the current reproduction number $R$ but also on the generation time distribution $g(s)$ (which describes the random time between getting infected and infecting others). The relation between $r$ and $R$ and $g(s)$ is given by the Euler–Lotka equation (e.g. [10]). If we make the pragmatic and common assumption that the generation time follows a Gamma distribution with mean $\mu$ and standard deviation $\sigma$, then the Euler–Lotka equation is given by

$$r = \frac{1}{\mu}c^{-2}\left(R^{c^2} - 1\right), \tag{A 1}$$

where $c = \sigma/\mu$ is the coefficient of variation of the generation time distribution ($c = 0$ is not possible since infections would then only happen at pre-specified time points). From this equation, it is seen that $r < 0$ (decline) if $R < 1$ and $r > 0$ (growth) if $R > 1$.

   The exponential growth/decline rate without face masks, $r_{No}$ should hence be computed with $R_{No}$ and the exponential growth/decline rate with mandatory face masks, $r_{FM}$, should be computed with $R_{FM}$. If we compare the growth rate with and without mandatory face masks it is seen that this relation will not only depend on the face mask effect $E_{FM} = 1 - R_{FM}/R_{No}$, but it will also depend on the value of the underlying reproduction number $R_{No}$ (and also on $\mu$ and $\sigma$). As a consequence, the effect of face masks on the exponential rate $r$ will not be the same in different communities, nor for the same community if comparing different time points of introducing face mask regulations if other preventive measures may have changed. When estimating $E_{FM}$ we will want to express $R$ in terms of the exponential growth rate. For this, we invert equation (A 1)

$$R = \left(c^2 r\mu + 1\right)^{c^{-2}}, \quad \text{where } c = \frac{\sigma}{\mu}. \tag{A 2}$$

## (c) Effect on change of incidence during a given time frame

The incidence $i(t)$ is defined as the number of new infections per day. In epidemic models, this incidence is assumed to be proportional to the current number of infectious individuals $I(t)$ (the more infectious people the higher the infection pressure). As a consequence, the ratio of the incidence at time $t + d$ and the incidence at $t$ will equal

$$\frac{i(t+d)}{i(t)} = \frac{I(t+d)}{I(t)} = e^{rd}, \quad \text{so } r = \frac{\ln[i(t+d)/i(t))]}{d}. \tag{A 3}$$

If the time frame $(t, t + d)$ has no face mask regulations then $r_{No}$ should be used, whereas $r_{FM}$ should be used if face masks are mandatory during this period. The face mask effect on the change of incidence-ratio will hence not only depend on $E_{FM}$ but also on the underlying reproduction number $R_{No}$. It follows that different communities typically will see different effects on incidence ratios, even when the same duration $d$ of time frames are used.

## (d) Effect on the number of infections during a given time frame

We now describe how the number of infections within a given time frame is affected by introduction of face mask regulation. If $N(t)$ denotes the number of infected individuals up to time $t$ we are hence interested in how $N(t + d) - N(t)$ changes depending on whether face masks are mandatory or not. This number depends on the exponential growth rate, but also on $I(t)$, the starting number of infectious individuals. As before we assume that the incidence $i(t) = N'(t)$ is proportional to $I(t)$: $i(t) = \lambda I(t)$, where $\lambda$ is the mean infection rate for infectives. From before we know that $I(t)$ grows exponentially so that $I(t + s) = I(t)e^{rs}$. As a consequence, we get the following expression for the (expected) number of infections during the studied time frame

$$N(t+d) - N(t) = \int_0^d N'(t+s)\,ds = \lambda I(t) \int_0^d e^{rs}\,ds = I(t)\frac{\lambda}{r}\left(e^{rd} - 1\right).$$

Putting observable quantities to one side we get $(N(t + d) - N(t))/I(t) = (\lambda/r)(e^{rd} - 1)$. This equation contains a new 'parameter' $\lambda$. In order to obtain a numerical value for $\lambda$, we make an additional approximation by relating $\lambda$ to the mean duration of the infectious period under the assumption that the underlying epidemic model is the Markovian SIR epidemic model (which assumes no latent period and an exponential infectious period). For this case $r = \lambda - \gamma$, where $1/\gamma$ equals the mean generation time earlier denoted as $\mu$. We hence have that $\lambda/r = 1 + 1/(\mu r)$, and we get

$$\frac{N(t+d) - N(t)}{I(t)} = \left(1 + \frac{1}{\mu r}\right)\left(e^{rd} - 1\right). \tag{A 4}$$

If the study period occurs when the community is not wearing face masks it should be evaluated with $r_{No}$, and using $r_{FM}$ if face masks are mandatory. Just like with the incidence ratio, the face mask effect on the number of infected not only depends on the starting number of infectives and the duration of the study period, but also on other community settings. This face mask effect is hence not transferable to other community settings either.

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
