## [Peer Review File · Proceedings. Mathematical, Physical, and Engineering Sciences]

Review History

RSPA-2021-0151.R0 (Original submission)

Review form: Referee 1

Is the manuscript an original and important contribution to its field?

Acceptable

Is the paper of sufficient general interest?

Good

Is the overall quality of the paper suitable?

Marginal

Can the paper be shortened without overall detriment to the main message?

No

Do you think some of the material would be more appropriate as an electronic appendix?

Yes

Do you have any ethical concerns with this paper?

No

Recommendation?

Major revision is needed (please make suggestions in comments)

Comments to the Author(s)

This paper looks at the topical problem of estimating the effect of wearing face masks and, specifically, the reduction in the (current) reproduction number, R , that results from the mandatory wearing of face masks as compared with no use of masks. While the theory set out in Sections 2 and 3 is straightforward and well-known, the application to observational data and interpretation of the results described in Sections 4 and 5 is of interest. The paper shows signs of having (understandably perhaps) been written in haste, and would benefit from careful proofreading as there are many typographical errors. In places the English lacks fluency and in general the descriptions could be expressed much more concisely and irrelevant details omitted (for example, why introduce notation for different sorts of vaccine efficacy in the Introduction?).

The effect of face mask wearing is to reduce the rate of effective contacts by lowering both susceptibility and transmission, and hence to reduce the reproduction number R by the same factor. The face mask effect EFM is defined in Section 2 to be the proportionate decrease in R . Most of the notation introduced in this section is not required. The equations in Section 3 express R in terms of various observable quantities such as the growth rate and increase in cases, thus enabling the estimation of EFM. These expressions could well be stated in an Appendix. There is no new theory here and the paper could itself be reduced to a short note, concentrating on Sections 4 and 5, without loss. The beginning of Section 5 would be a suitable introduction.

The estimation and discussion sections (Section 4 and 5) demonstrate the calculation of EFM using various data sets, and have some interesting and potentially useful results, although these are necessarily qualified by all sorts of caveats. Mandatory face-covering does not guarantee compliance and the latter may vary substantially between communities. More importantly for the work presented here, the connections between R and the various observable quantities taken from Section 3 are for an emerging epidemic in a homogeneously mixing population and even, in 3.3 for a Markov SIR model, and it is assumed that everything in terms of mixing behaviour and other interventions remains fixed during the period over which the effect of face masks is to be estimated. The sensitivity of the results to all these assumptions needs to be discussed. There is some discussion of confounding effects but it seems unlikely that individual behaviour would have remained unchanged in the face of a rapidly growing pandemic during the time periods covered by these studies. How is population heterogeneity likely to affect the results? Is it reasonable to apply theory derived for an emerging epidemic to data from Kansas in July 2020?

Minor point:

Equation 2 would look simpler in terms of the index of dispersion of the generation time distribution rather than its coefficient of variation

Review form: Referee 2

Is the manuscript an original and important contribution to its field?

Good

Is the paper of sufficient general interest?

Good

Is the overall quality of the paper suitable?

Good

Can the paper be shortened without overall detriment to the main message?

No

Do you think some of the material would be more appropriate as an electronic appendix?

Yes

Do you have any ethical concerns with this paper?

Yes

Recommendation?

Accept with minor revision (please list in comments)

Comments to the Author(s)

Major

1. I may be wrong but much of the theory in section 3 seems applicable to any intervention that may impact on both infectivity and susceptibility, including vaccination.
2. Section 2 is important, showing how face mask effects impinge on reproduction number. It is less clear what section 3 adds, which is largely standard theory on how the reproduction number relates to other epidemiological parameters.
3. A limitation of the data from the Kansas study is that compliance would not have been 100% in counties with mandatory face masks, nor 0% in counties without mandatory face masks. It is therefore difficult to assess individual transmission parameters.
4. The description of the paper in section 4.4 is very brief, in contrast to 4.1, 4.2 and 4.3. This makes the paper feel unbalanced. Is this study reliable?
5. Is it possible to derive confidence intervals for the studies in 4.2 and 4.3? How dependable are these findings?
6. The authors might like to mention the possibility of publication bias i.e. only positive findings making it to publication.

Minor

1. Some of the nomenclature is questionable. 4.1 is not a case-control study. 4.2 and 4.3 not retrospective studies.
2. In section 4.3 the incidence of 42 should be 142?

Decision letter (RSPA-2021-0151.R0)

14-Apr-2021

Dear Dr Britton

The Editor of Proceedings A has now received comments from referees on the above paper and would like you to revise it in accordance with their suggestions which can be found below (not including confidential reports to the Editor).

Please submit a copy of your revised paper within four weeks - if we do not hear from you within this time then it will be assumed that the paper has been withdrawn. In exceptional circumstances, extensions may be possible if agreed with the Editorial Office in advance.

Please note that it is the editorial policy of Proceedings A to offer authors one round of revision in which to address changes requested by referees. If the revisions are not considered satisfactory by the Editor, then the paper will be rejected, and not considered further for publication by the journal. In the event that the author chooses not to address a referee's comments, and no scientific justification is included in their cover letter for this omission, it is at the discretion of the Editor whether to continue considering the manuscript.

To revise your manuscript, log into <http://mc.manuscriptcentral.com/prsa> and enter your Author Centre, where you will find your manuscript title listed under "Manuscripts with Decisions." Under "Actions," click on "Create a Revision." Your manuscript number has been appended to denote a revision.

You will be unable to make your revisions on the originally submitted version of the manuscript. Instead, revise your manuscript and upload a new version through your Author Centre.

When submitting your revised manuscript, you will be able to respond to the comments made by the referee(s) and upload a file "Response to Referees" in Step 1: "View and Respond to Decision Letter". Please use this to document how you have responded to the comments, and the adjustments you have made. In order to expedite the processing of the revised manuscript, please be as specific as possible in your response to the referee(s).

IMPORTANT: Your original files are available to you when you upload your revised manuscript. Please delete any unnecessary previous files before uploading your revised version.

When revising your paper please ensure that it remains under 28 pages long. In addition, any pages over 20 will be subject to a charge (£150 + VAT (where applicable) per page). Your paper has been ESTIMATED to be 9 pages.

Open Access

You are invited to opt for open access, our author pays publishing model. Payment of open access fees will enable your article to be made freely available via the Royal Society website as soon as it is ready for publication. For more information about open access please visit <https://royalsociety.org/journals/authors/open-access/>. The open access fee for this journal is £1700/\$2380/€2040 per article. VAT will be charged where applicable. Please note that if the corresponding author is at an institution that is part of a Read and Publishing deal you are required to select this option. See <https://royalsociety.org/journals/librarians/purchasing/read-and-publish/read-publish-agreements/> for further details.

Once again, thank you for submitting your manuscript to Proc. R. Soc. A and I look forward to receiving your revision. If you have any questions at all, please do not hesitate to get in touch.

Yours sincerely
Raminder Shergill
proceedingsa@royalsociety.org

on behalf of
Professor Vincenzo Capasso
Board Member
Proceedings A

Reviewer(s)' Comments to Author:

Referee: 1

Comments to the Author(s)

This paper looks at the topical problem of estimating the effect of wearing face masks and, specifically, the reduction in the (current) reproduction number, R , that results from the mandatory wearing of face masks as compared with no use of masks. While the theory set out in Sections 2 and 3 is straightforward and well-known, the application to observational data and interpretation of the results described in Sections 4 and 5 is of interest. The paper shows signs of having (understandably perhaps) been written in haste, and would benefit from careful proofreading as there are many typographical errors. In places the English lacks fluency and in general the descriptions could be expressed much more concisely and irrelevant details omitted (for example, why introduce notation for different sorts of vaccine efficacy in the Introduction?).

The effect of face mask wearing is to reduce the rate of effective contacts by lowering both susceptibility and transmission, and hence to reduce the reproduction number R by the same factor. The face mask effect EFM is defined in Section 2 to be the proportionate decrease in R . Most of the notation introduced in this section is not required. The equations in Section 3 express R in terms of various observable quantities such as the growth rate and increase in cases, thus enabling the estimation of EFM. These expressions could well be stated in an Appendix. There is no new theory here and the paper could itself be reduced to a short note, concentrating on Sections 4 and 5, without loss. The beginning of Section 5 would be a suitable introduction.

The estimation and discussion sections (Section 4 and 5) demonstrate the calculation of EFM using various data sets, and have some interesting and potentially useful results, although these are necessarily qualified by all sorts of caveats. Mandatory face-covering does not guarantee compliance and the latter may vary substantially between communities. More importantly for the work presented here, the connections between R and the various observable quantities taken from Section 3 are for an emerging epidemic in a homogeneously mixing population and even, in 3.3 for a Markov SIR model, and it is assumed that everything in terms of mixing behaviour and other interventions remains fixed during the period over which the effect of face masks is to be estimated. The sensitivity of the results to all these assumptions needs to be discussed. There is some discussion of confounding effects but it seems unlikely that individual behaviour would have remained unchanged in the face of a rapidly growing pandemic during the time periods covered by these studies. How is population heterogeneity likely to affect the results? Is it reasonable to apply theory derived for an emerging epidemic to data from Kansas in July 2020?

Minor point:

Equation 2 would look simpler in terms of the index of dispersion of the generation time distribution rather than its coefficient of variation

Referee: 2

Comments to the Author(s)

Major

1. I may be wrong but much of the theory in section 3 seems applicable to any intervention that may impact on both infectivity and susceptibility, including vaccination.
2. Section 2 is important, showing how face mask effects impinge on reproduction number. It is less clear what section 3 adds, which is largely standard theory on how the reproduction number relates to other epidemiological parameters.
3. A limitation of the data from the Kansas study is that compliance would not have been 100% in counties with mandatory face masks, nor 0% in counties without mandatory face masks. It is therefore difficult to assess individual transmission parameters.

4. The description of the paper in section 4.4 is very brief, in contrast to 4.1, 4.2 and 4.3. This makes the paper feel unbalanced. Is this study reliable?

5. Is it possible to derive confidence intervals for the studies in 4.2 and 4.3? How dependable are these findings?

6. The authors might like to mention the possibility of publication bias i.e. only positive findings making it to publication.

Minor

1. Some of the nomenclature is questionable. 4.1 is not a case-control study. 4.2 and 4.3 not retrospective studies.

2. In section 4.3 the incidence of 42 should be 142?

Board Member:

Comments to Author(s):

Based on the comments by the two Reviewers, the authors are welcome to submit a thoroughly revised manuscript.

Please additionally submit a letter of responses to the comments, one by one.

Author's Response to Decision Letter for (RSPA-2021-0151.R0)

See Appendix A.

RSPA-2021-0151.R1 (Revision)

Review form: Referee 1

Is the manuscript an original and important contribution to its field?

Good

Is the paper of sufficient general interest?

Good

Is the overall quality of the paper suitable?

Good

Can the paper be shortened without overall detriment to the main message?

No

Do you think some of the material would be more appropriate as an electronic appendix?

Yes

Do you have any ethical concerns with this paper?

No

Recommendation?

Accept with minor revision (please list in comments)

Comments to the Author(s)

The extra comments and discussion have usefully addressed the issues that were raised by the reviewers. I have no further comments.

Review form: Referee 2

Is the manuscript an original and important contribution to its field?

Good

Is the paper of sufficient general interest?

Good

Is the overall quality of the paper suitable?

Good

Can the paper be shortened without overall detriment to the main message?

Yes

Do you think some of the material would be more appropriate as an electronic appendix?

No

Do you have any ethical concerns with this paper?

No

Recommendation?

Accept as is

Comments to the Author(s)

No comments

Decision letter (RSPA-2021-0151.R1)

13-May-2021

Dear Dr Britton,

On behalf of the Editor, I am pleased to inform you that your Manuscript RSPA-2021-0151.R1 entitled "Quantifying the preventive effect of wearing face masks" has been accepted for publication subject to minor revisions in Proceedings A. Please find the referees' comments below.

The reviewer(s) have recommended publication, but also suggest some minor revisions to your manuscript. Therefore, I invite you to respond to the reviewer(s)' comments and revise your manuscript. Please note that we have a strict upper limit of 28 pages for each paper. Please endeavour to incorporate any revisions while keeping the paper within journal limits. Please note that page charges are made on all papers longer than 20 pages. If you cannot pay these charges you must reduce your paper to 20 pages before submitting your revision. Your paper has

been ESTIMATED to be 12 pages. We cannot proceed with typesetting your paper without your agreement to meet page charges in full should the paper exceed 20 pages when typeset. If you have any questions, please do get in touch.

It is a condition of publication that you submit the revised version of your manuscript within 7 days. If you do not think you will be able to meet this date please let me know in advance of the due date.

To revise your manuscript, log into <https://mc.manuscriptcentral.com/prsa> and enter your Author Centre, where you will find your manuscript title listed under "Manuscripts with Decisions." Under "Actions," click on "Create a Revision." Your manuscript number has been appended to denote a revision.

You will be unable to make your revisions on the originally submitted version of the manuscript. Instead, revise your manuscript and upload a new version through your Author Centre.

When submitting your revised manuscript, you will be able to respond to the comments made by the referee(s) and upload a file "Response to Referees" in Step 1: "View and Respond to Decision Letter". You can use this to document any changes you make to the original manuscript. In order to expedite the processing of the revised manuscript, please be as specific as possible in your response to the referee(s).

IMPORTANT: Your original files are available to you when you upload your revised manuscript. Please delete any redundant files before completing the submission process.

When uploading your revised files, please make sure that you include the following as we cannot proceed without these:

- 1) A text file of the manuscript (doc, txt, rtf or tex), including the references, tables (including captions) and figure captions. Please remove any tracked changes from the text before submission. PDF files are not an accepted format for the "Main Document".
- 2) A separate electronic file of each figure (tif, eps or print-quality pdf preferred). The format should be produced directly from original creation package, or original software format.
- 3) Electronic Supplementary Material (ESM): all supplementary materials accompanying an accepted article will be treated as in their final form. Note that the Royal Society will not edit or typeset supplementary material and it will be hosted as provided. Please ensure that the supplementary material includes the paper details where possible (authors, article title, journal name). Supplementary files will be published alongside the paper on the journal website and posted on the online figshare repository (<https://figshare.com>). The heading and legend provided for each supplementary file during the submission process will be used to create the figshare page, so please ensure these are accurate and informative so that your files can be found in searches. Files on figshare will be made available approximately one week before the accompanying article so that the supplementary material can be attributed a unique DOI. Alternatively you may upload a zip folder containing all source files for your manuscript as described above with a PDF as your "Main Document". This should be the full paper as it appears when compiled from the individual files supplied in the zip folder.

Article Funder

Please ensure you fill in the Article Funder question on page 2 to ensure the correct data is collected for FundRef (<http://www.crossref.org/fundref/>).

Media summary

Please ensure you include a short non-technical summary (up to 100 words) of the key findings/ importance of your paper. This will be used for to promote your work and marketing purposes (e.g. press releases). The summary should be prepared using the following guidelines:

- *Write simple English: this is intended for the general public. Please explain any essential technical terms in a short and simple manner.
- *Describe (a) the study (b) its key findings and (c) its implications.
- *State why this work is newsworthy, be concise and do not overstate (true 'breakthroughs' are a rarity).
- *Ensure that you include valid contact details for the lead author (institutional address, email address, telephone number).

Cover images

We welcome submissions of images for possible use on the cover of Proceedings A. Images should be square in dimension and please ensure that you obtain all relevant copyright permissions before submitting the image to us. If you would like to submit an image for consideration please send your image to proceedingsa@royalsociety.org

Open Access

You are invited to opt for open access, our author pays publishing model. Payment of open access fees will enable your article to be made freely available via the Royal Society website as soon as it is ready for publication. For more information about open access please visit <https://royalsociety.org/journals/authors/open-access/>. The open access fee for this journal is £1700/\$2380/€2040 per article. VAT will be charged where applicable. Please note that if the corresponding author is at an institution that is part of a Read and Publishing deal you are required to select this option. See <https://royalsociety.org/journals/librarians/purchasing/read-and-publish/read-publish-agreements/> for further details.

Once again, thank you for submitting your manuscript to Proceedings A and I look forward to receiving your revision. If you have any questions at all, please do not hesitate to get in touch.

Best wishes
Raminder Shergill
proceedingsa@royalsociety.org
Proceedings A

on behalf of
Professor Vincenzo Capasso
Board Member
Proceedings A

Reviewer(s)' Comments to Author:

Referee: 1

Comments to the Author(s)

The extra comments and discussion have usefully addressed the issues that were raised by the reviewers. I have no further comments.

Referee: 2

Comments to the Author(s)

No comments

Board Member

Comments to Author(s):

Based on the comments by the two Reviewers the paper is acceptable.

It is advisable that Section 3 be moved to an Appendix following the main text. Otherwise the Supplementary Material would be a less desirable possibility.

Decision letter (RSPA-2021-0151.R2)

10-Jun-2021

Dear Dr Britton

I am pleased to inform you that your manuscript entitled "Quantifying the preventive effect of wearing face masks" has been accepted in its final form for publication in Proceedings A.

Our Production Office will be in contact with you in due course. You can expect to receive a proof of your article soon. Please contact the office to let us know if you are likely to be away from e-mail in the near future. If you do not notify us and comments are not received within 5 days of sending the proof, we may publish the paper as it stands.

COVID-19 rapid publication process: We are taking steps to expedite the publication of research relevant to the pandemic. If you wish, you can opt to have your paper published as soon as it is ready, rather than waiting for it to be published on the scheduled Wednesday. This means your paper will not be included in the weekly media round-up which the Society sends to journalists ahead of publication. However, it will appear in the COVID-19 Publishing Collection which journalists will be directed to each week (<https://royalsocietypublishing.org/topic/special-collections/novel-coronavirus-outbreak>) If you wish to have your paper published immediately please notify proca_proofs@royalsociety.org and press@royalsociety.org

The Royal Society has signed a Wellcome statement on the subject of research findings and data relevant to the coronavirus (COVID-19) outbreak. We are one of several signatories to this statement and our collective aim is to ensure that the relevant research and data are shared rapidly and openly in order to inform the worldwide public health response and to help save lives. We are therefore making papers related to COVID-19 open access free of charge.

As a reminder, you have provided the following 'Data accessibility statement' (if applicable). Please remember to make any data sets live prior to publication, and update any links as needed when you receive a proof to check. It is good practice to also add data sets to your reference list.
Statement (if applicable):

Under the terms of our licence to publish you may post the author generated postprint (ie. your accepted version not the final typeset version) of your manuscript at any time and this can be made freely available. Postprints can be deposited on a personal or institutional website, or a recognised server/repository. Please note however, that the reporting of postprints is subject to a media embargo, and that the status the manuscript should be made clear. Upon publication of the definitive version on the publisher's site, full details and a link should be added.

You can cite the article in advance of publication using its DOI. The DOI will take the form: 10.1098/rspa.XXXX.YYYY, where XXXX and YYYY are the last 8 digits of your manuscript number (eg. if your manuscript number is RSPA-2017-1234 the DOI would be 10.1098/rspa.2017.1234).

For tips on promoting your accepted paper see our blog post:
<https://royalsociety.org/blog/2020/07/promoting-your-latest-paper-and-tracking-your-results/>

On behalf of the Editor of Proceedings A, we look forward to your continued contributions to the Journal.

Sincerely,
Raminder Shergill
proceedingsa@royalsociety.org

Appendix A

Response to comments of the referees and Associate editor

First I would like to thank the two referees and the Associate for many useful comments which helped improve the manuscript. The manuscript has been revised accordingly, also correcting a few typos and other errors.

Please find below all comments by the referees and my response to each of them. In order to distinguish between referee's comments and my response, my responses are written in italics and start with ">>".

Reviewer(s)' Comments to Author:

Referee: 1

Comments to the Author(s)

This paper looks at the topical problem of estimating the effect of wearing face masks and, specifically, the reduction in the (current) reproduction number, R , that results from the mandatory wearing of face masks as compared with no use of masks. While the theory set out in Sections 2 and 3 is straightforward and well-known, the application to observational data and interpretation of the results described in Sections 4 and 5 is of interest. The paper shows signs of having (understandably perhaps) been written in haste, and would benefit from careful proofreading as there are many typographical errors. In places the English lacks fluency and in general the descriptions could be expressed much more concisely and irrelevant details omitted (for example, why introduce notation for different sorts of vaccine efficacy in the Introduction?).

>> The manuscript has now been carefully edited removing several typos and other inconsistencies. Further, unnecessary technical parts in Section 1 and 3 have been removed.

The effect of face mask wearing is to reduce the rate of effective contacts by lowering both susceptibility and transmission, and hence to reduce the reproduction number R by the same factor. The face mask effect EFM is defined in Section 2 to be the proportionate decrease in R . Most of the notation introduced in this section is not required. The equations in Section 3 express R in terms of various observable quantities such as the growth rate and increase in cases, thus enabling the estimation of EFM. These expressions could well be stated in an Appendix. There is no new theory here and the paper could itself be reduced to a short note, concentrating on Sections 4 and 5, without loss. The beginning of Section 5 would be a suitable introduction.

>> Many good points! Unnecessary notation in Section 2 is dropped. Some notation and equations in Section 3 have also been dropped. Instead the initial discussion on under what assumptions the modelling results are valid has been extended upon request (new Section 3.1). It is true that the theory presented in Section 3 does not contain any new theory. However, finding the different results in the literature

requires quite a lot of searching and I think it is worth gathering them in one place. I checked several other manuscripts in Roy Soc Proc A and found that many of them contained a section called “Mathematical background” (which the section now is renamed to, plus “assumptions”). I did not find any papers having an appendix in the main text and would prefer not putting this theory in a Supplementary Material which readers may not find. If allowed by the journal I can move Section 3 into an Appendix at the end of the main text, or if requested to a separate Supplementary Material.

The estimation and discussion sections (Section 4 and 5) demonstrate the calculation of EFM using various data sets, and have some interesting and potentially useful results, although these are necessarily qualified by all sorts of caveats. Mandatory face-covering does not guarantee compliance and the latter may vary substantially between communities. More importantly for the work presented here, the connections between R and the various observable quantities taken from Section 3 are for an emerging epidemic in a homogeneously mixing population and even, in 3.3 for a Markov SIR model, and it is assumed that everything in terms of mixing behaviour and other interventions remains fixed during the period over which the effect of face masks is to be estimated. The sensitivity of the results to all these assumptions needs to be discussed. There is some discussion of confounding effects but it seems unlikely that individual behaviour would have remained unchanged in the face of a rapidly growing pandemic during the time periods covered by these studies. How is population heterogeneity likely to affect the results? Is it reasonable to apply theory derived for an emerging epidemic to data from Kansas in July 2020?

>> Many important points! I have added a new Section 3.1 detailing the underlying assumptions. The Conclusion/Discussion section has been expanded with additional paragraphs. In 2nd full paragraph of p10 I discuss effects of confounding errors due to differences and changes during the study period. In the next paragraph effects of possible publication bias is brought up. In the paragraph thereafter I discuss effects of adherence and also how changed behaviour during the study period may affect estimation.

Minor point:

Equation 2 would look simpler in terms of the index of dispersion of the generation time distribution rather than its coefficient of variation

>> Index of dispersion is defined by σ^2/μ . I agree with the comment if it would have been defined as σ^2/μ^2 , but not now. Further, I prefer coefficient of variation since it lacks measurement unit. For this reason I did not change to dispersion index.

Referee: 2

Comments to the Author(s)

Major

1. I may be wrong but much of the theory in section 3 seems applicable to any

intervention that may impact on both infectivity and susceptibility, including vaccination.

>> *Correct. This is now clarified (end of Sec 2).*

2. Section 2 is important, showing how face mask effects impinge on reproduction number. It is less clear what section 3 adds, which is largely standard theory on how the reproduction number relates to other epidemiological parameters.

>> *See my second comment to Referee 1. In short: I reduced formulae, inserted some discussion about assumptions but would prefer to keep it in main text. I am happy to move it to an Appendix in main text if possible.*

3. A limitation of the data from the Kansas study is that compliance would not have been 100% in counties with mandatory face masks, nor 0% in counties without mandatory face masks. It is therefore difficult to assess individual transmission parameters.

>> *Correct. This is discussed at bottom of p10.*

4. The description of the paper in section 4.4 is very brief, in contrast to 4.1, 4.2 and 4.3. This makes the paper feel unbalanced. Is this study reliable?

>> *I have added a few more lines of discussion, including short-comings. The main reason why 4.4 is shorter is however that that study estimates EFM, so no conversions are needed. Since I have found four empirical studies that estimate face mask effects I prefer not to leave anyone out.*

5. Is it possible to derive confidence intervals for the studies in 4.2 and 4.3? How dependable are these findings?

>> *In the second full paragraph of p10 I discuss potential errors in all studies. The Danish study has high uncertainty whereas the studies in 4.2 and 4.3 have much less uncertainty but on the other hand higher risk of biases. Producing confidence intervals for these studies would require substantial work and method development, and since the possible biases are more important I have not pursued this task.*

6. The authors might like to mention the possibility of publication bias i.e. only positive findings making it to publication.

>> *Good comment. Now commented in penultimate paragraph of p10.*

Minor

1. Some of the nomenclature is questionable. 4.1 is not a case-control study. 4.2 and 4.3 not retrospective studies.

>> *Thanks. Corrected.*

2. In section 4.3 the incidence of 42 should be 142?

>> *Thanks. Corrected.*

Board Member:

Comments to Author(s):

Based on the comments by the two Reviewers, the authors are welcome to submit a thoroughly revised manuscript.

Please additionally submit a letter of responses to the comments, one by one.

>> *Thanks. Above I comment how all the referee comments have been addressed in the revised manuscript.*